# ONE-VERSUS-OTHERS ATTENTION: SCALABLE MULTIMODAL INTEGRATION

## ABSTRACT

Multimodal learning models have become increasingly important as they surpass single-modality approaches on diverse tasks ranging from question-answering to autonomous driving. Despite the importance of multimodal learning, existing efforts focus on NLP applications, where the number of modalities is typically at most four (images, text, audio, video). However, data inputs in other domains, such as clinical medicine, may include many more modalities like X-rays, PET scans, MRIs, genetic screening, genomic data, clinical notes, creating a need for both efficient and accurate information fusion. Many state-of-the-art models rely on pairwise cross-attention or early fusion through self-attention, which do not scale well for applications with more than three modalities. The complexity per layer of computing attention in either paradigm is, at best, quadratic with the number of modalities, potentially requiring considerable computational resources. To address this, we propose a new attention mechanism, One-Versus-Others (OvO) attention, that scales linearly with the number of modalities, thus offering a significant reduction in computational complexity compared to existing multimodal attention methods. Using three diverse real-world datasets as well as an additional simulation experiment, we show that our method improves performance compared to popular fusion techniques while decreasing computation costs [1].

## 1 INTRODUCTION

Multimodal learning has emerged as a promising approach, which enables joint learning from multiple modalities of data (e.g., text and images). Combining different modalities allows for a more comprehensive and accurate understanding of tasks such as image and video captioning (46; 33), audio-visual speech recognition (36), sentiment analysis (29), and medical decision support (21). Multimodal learning has been explored through various backbone methods in machine learning (e.g., Decision Trees) and various Neural Network architectures. While feature-level integration was mostly used in more traditional machine learning algorithms, Neural Networks have allowed for the intermediate fusion of modalities through layers and late fusion at the decision-making stage. However, both fusion paradigms lack a key component - capturing explicit interaction between modalities. For example, in detecting hateful social media posts (41), imaging features help reinforce and ground the textual information and thus lead to more robust decision-making. The success of the Transformer (43) on various NLP tasks and, consequently, the Vision Transformer's (ViT (8)) success on vision tasks motivated the extension of the Transformer to the multimodal case. Multimodal Transformers, such as LXMERT (40) and ViLBERT (19), introduced a fusion method that captures interactions between modalities using *cross-attention*. On the other hand, models such as VisualBERT (16) and VL-BERT (37) used early fusion, where vision and language inputs are concatenated early to learn multimodal through *self-attention*. However, most multimodal models are based on vision-language tasks, requiring a one-to-one correspondence between all modalities. For example, such an alignment may temporally sync a video, the corresponding audio track, and a textual transcript of people talking. Facial movement, sound waves, and textual inputs describe and complement each other. In other scenarios, such as clinical decision support, data inputs may include an MRI scan and genetic screening to diagnose a disease. While both are valid multimodal inputs, even domain experts cannot tell which pixels in the image correspond to a specific mutation. Thus, vision-language models that rely on modality alignment are unsuitable for generalized multimodal applications, motivating the need for a domain-neutral approach.

For domains producing datasets without alignment, self-attention or cross-attention remain viable strategies (10) for data integration. However, both self-attention and cross-attention grow quadratically

---

[1]Code is available at `https://anonymous.4open.science/r/OvO-09E6/`

with the number of modalities, posing a scalability challenge. To address this gap, we propose a new attention mechanism, One-Versus-Others (OvO) attention. OvO takes outputs from *one* modality encoder and computes the dot product against a weight matrix and the average of the weights from all *other* modalities encoders (hence the name, One-Versus-Others). Our approach significantly reduces computational complexity, as it grows linearly with the number of modalities (see Section 3.3). Figure 1 shows a four-modality example to demonstrate the difference between our approach (scales linearly) and self-attention/cross-attention (scales quadratically). OvO is a general attention scheme that can be integrated into existing multimodal architectures instead of cross-attention or self-attention. We validated our approach on a simulation dataset showing scalability gains in an extreme multimodal setting (20 modalities). Furthermore, we used three diverse real-world datasets that vary in modalities, encoder types (pre-trained and not), number of samples, and application domains to show our model's versatility in different multimodal settings. Our results demonstrate that our method improves performance compared to self-attention and cross-attention while decreasing computation costs. Concretely, we make the following contributions: (1) we present, OvO, a generalizable multimodal integration scheme that is domain-neutral and does not require modality alignment; (2) OvO scales linearly with the number of modalities while also performing competitively to self-attention and cross-attention; (3) we perform robust benchmarking on new simulated and real-world multimodal integration tasks.

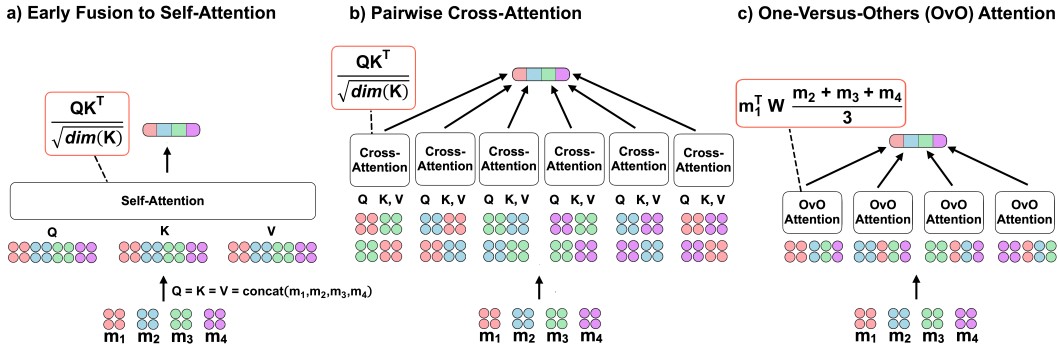

Figure 1: **Integration scheme comparison**. (a) Early fusion to self-attention with scaled dot product attention (43), and (b) Pairwise cross-attention integration with scaled dot product attention (43). (b) Our proposed method, One-Versus-Others (OvO), does not rely on pairwise interactions or long concatenated sequences but rather captures all modalities in a single attention score. A modality is represented by $m_i$ and $W$ is a learnable parameter (see Section 3.1).

## 2 RELATED WORK

Commonly, the representations from each modality in the multimodal Transformers are passed through one of two paradigms - early fusion followed by self-attention or fusion through cross-attention. The early fusion group (e.g., Uniter (4), VisualBERT (16), Vl-BERT (37)) extends the BERT architecture by concatenating the embedded visual inputs and the embedded textual inputs as a single input, before passing the inputs through attention (see Figure 1 (a)). Given modalities $m_1$ and $m_2$, queries $(Q)$, keys $(K)$, and values $(V)$ are computed from their concatenated sequence (e.g., $Q_{1,2} = concat(m_1, m_2)$). The final output from a standard Transformer block is denoted by Z, Equation 1 shows the early fusion paradigm.

$$\begin{cases} \text{Z}_{1,2} = MultiheadedAttention\left(\text{Q}_{1,2}, \text{K}_{1,2}, \text{V}_{1,2}\right) \\ \text{Z} = Transformer(Z_{1,2}) \end{cases} \tag{1}$$

The cross-attention scheme (used in ViLBERT (19), LXMERT (40), ActBERT (47), MulT (42), etc.) inputs each modality into its own Transformer, the outputs of which are fed to a cross-modal Transformer (see Figure 1 (b)). For such Transformers, the cross-modal interactions are captured through cross-attention, where queries $(Q)$, keys $(K)$, and values $(V)$ are computed from the modality inputs ($m_1$ and $m_2$), and then the keys and values from each modality are fed to the multi-headed

attention block of the other modality. The output, $Z$, is shown in Equation 2.

$$\begin{cases} Z_1 = MultiheadedAttention\left(Q_2, K_1, V_1\right) \\ Z_2 = MultiheadedAttention\left(Q_1, K_2, V_2\right) \\ Z = Transformer\left(concat\left(Z_1, Z_2\right)\right) \end{cases} \quad (2)$$

Since many multimodal tasks are centered in natural language (video and image captioning, visual question and answering, audio-visual speech recognition, image retrieval, etc.), it is no surprise that BERT (6) is a key component of many multimodal Transformers (e.g., VideoBERT (38), ViLBERT (19), VisualBERT (16), VL-BERT (37), Pixel-BERT (13), ActBERT (47), ImageBERT (30)). However, most concrete innovations have been focused on improving performance for specific natural language tasks rather than building new domain-neutral multimodal integration methods. While the early fusion and cross-attention paradigms could be extended to three modalities, seen in TriBERT (31) and VATT (1), these models face scalability challenges for more than three modalities. Cross-attention methods can leverage joint representations formed from cross-attention but do not scale well to larger numbers of modalities as they are computed in a pairwise fashion. Thus, if there are $k$ modalities, computing pairwise fusion between each pair will result in $\binom{k}{2}$ matrix computations. Moreover, attention is not a symmetric calculation, which means that most commonly, it is computed bi-directionally (e.g., image to text and text to image), leading to an even greater computational burden. Early fusion involves the concatenation of modalities before the Transformer layer, which similarly does not scale well with the number of modalities. Self-attention is quadratic with respect to sequence length (43), and since early fusion methods concatenate inputs before attention, the computational complexity will increase quadratically as the number of modalities increases (see Section 3.3). Furthermore, concatenation is not order invariant, making the ordering of modalities an important consideration, potentially requiring similar bi-directional computations as cross-attention. Our integration method, OvO, addresses the limitations mentioned above in a scalable and domain-neutral manner.

## 3 METHODS

### 3.1 ONE-VERSUS-OTHERS (OVO) ATTENTION

Instead of integrating modalities in a pairwise way or through a concatenated stream, we propose a new attention mechanism, One-Versus-Others (OvO) Attention, which grows linearly with number of modalities rather than quadratically as required by cross-attention or self-attention (see Section A). Given modality $m_i$, where $k$ is the number of modalities and $i \in \{1, 2, \ldots, k\}$, $W$ is a neural network weight matrix that is shared across all modalities (see Figure 1 (c)). The similarity score function calculated for modality $m_i$ with respect to a set of other modalities ($m_j : j \neq i$) is shown in Equation 3, and the context vector for modality $m_i$ using OvO Attention is shown in Equation 4:

$$score\left(m_i, \{m_j : j \neq i\}\right) = m_i^T W \frac{\sum_{j \neq i}^{k} m_j}{k - 1} \quad (3)$$

$$OvOAttention\left(m_i, \{m_j : j \neq i\}\right) = softmax(score\left(m_i, \{m_j : j \neq i\}\right)) \cdot m_i \quad (4)$$

The formula takes in one modality and computes the dot product against all the other modalities with a weight matrix that can learn interactions throughout training. We chose to sum over the "other" modalities instead of concatenation for two reasons: (1) the concatenation vector will continue to increase in length with the number of modalities, which will result in a less scalable framework; (2) concatenation is not invariant to the order of modalities, which could affect the model prediction, whereas a sum provides position invariance.

Note that for $k = 2$ modalities ($m_1, m_2$), the similarity score function simplifies to that of general attention (20):

$$score\left(m_1, \{m_2\}\right) = m_1^T W \frac{m_2}{1} = m_1^T W m_2 \quad (5)$$

## 3.2 MULTI-HEADED OvO ATTENTION

To directly compare early fusion through self-attention and pairwise cross-attention, we extend OvO attention to the multi-headed attention framework. Multi-headed attention allows the model to attend to different input subspaces simultaneously. This is achieved by splitting the input embeddings into multiple linear projections, each processed independently through a self-attention mechanism. The outputs of each attention head are then concatenated and projected again to obtain the final output of the multi-headed attention layer. Formally, taking the input modality $m_i$ with respect to a set of other modalities $(m_j : j \neq i)$, the multi-headed attention layer for OvO attention is defined as follows:

$$\begin{cases} MultiheadedOvOAttention(m_i, \{m_j : j \neq i\}) = concat(head_1, \ldots, head_h)W^O \\ head_k = OvOAttention(m_i W_k^{m_i}, \{m_j W_k^{m_j} : j \neq i\}) \end{cases} \quad (6)$$

Here, $h$ is the number of attention heads, $W_k$ is a learnable weight matrix for the $k$-th attention head, $W^O$ is a learnable weight matrix that projects the concatenated outputs of the attention heads back to the original dimension, and OvO Attention is defined in Equation 4.

## 3.3 MODEL COMPLEXITIES

This section highlights the complexities associated with the three paradigms used in our experimental setting: early fusion followed by self-attention, pairwise cross-attention, and One-Versus-Others (OvO) Attention. Table 1 summarizes the complexity per layer. Let $k$ represent the number of modalities, $n$ be the feature length of each modality (assuming equal), and $d$ be the representation dimension of the respective weight matrices. As established in (43), self-attention has complexity of $\mathcal{O}(n^2 \cdot d)$. In the multimodal case, self-attention concatenates modalities before attention, leading to a sequence length of $k \cdot n$, influencing the quadratic term. Thus, the complexity of self-attention is $\mathcal{O}((k \cdot n)^2 \cdot d) = \mathcal{O}(k^2 \cdot n^2 \cdot d)$. Cross-attention computes attention over all pairwise permutations of modalities: $_kP_2 = \frac{k!}{(k-2)!} = k(k-1)$. Thus, the number of operations required by cross-attention is $\mathcal{O}(k \cdot (k-1) \cdot n^2 \cdot d) = \mathcal{O}((k^2 - k) \cdot n^2 \cdot d)$. When focusing on the fastest-growing terms in big $O$ notation, the final complexity per layer simplifies to $\mathcal{O}(k^2 \cdot n^2 \cdot d)$. One-Versus-Others (OvO) Attention requires one attention calculation per modality, making it linear with respect to $k$. Thus, the complexity per layer for OvO is $\mathcal{O}(k \cdot n^2 \cdot d)$. Appendix Section A provides step-by-step details for the complexity calculations.

Table 1: **Per-Layer complexities of model paradigm.** The per-layer complexity of early fusion through self-attention, cross-attention, and OvO attention are shown as a function of number of modalities $k$, feature-length of a modality $n$, and representation dimension $d$.

| Model | Complexity Per Layer |
|---|---|
| Self-Attention | $\mathcal{O}(k^2 \cdot n^2 \cdot d)$ |
| Cross-Attention | $\mathcal{O}(k^2 \cdot n^2 \cdot d)$ |
| One-Versus-Others (OvO) Attention | $\mathcal{O}(k \cdot n^2 \cdot d)$ |

## 4 EXPERIMENTS

We used one simulation experiment and three diverse real-world datasets to examine our method against three standard integration techniques: concatenation with no attention (baseline), early fusion with self-attention, and cross-attention. Note that even though our method is only computationally beneficial when combining three or more modalities, we used a two-modality dataset as a base-case to ensure that we are not compromising performance in any multimodal scenario, even where we do not offer scalability gains. We chose datasets where modality alignment is not apparent as such tasks are less frequently explored but still essential to solve (e.g., clinical decision support).

## 4.1 SIMULATION DATASET

We simulated 20 modalities under an artificial constraint that all modalities are needed to obtain an accurate classification. This is, for example, analogous to a medical setting where a physician

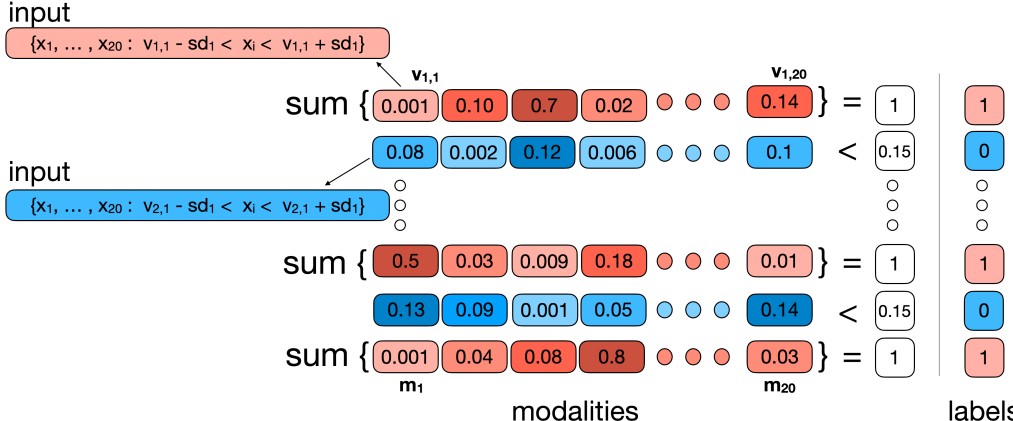

Figure 2: **Overview of the simulation dataset**. We simulated 20 modalities to test the capabilities of our method. The first classification label was created by 20 random values that add up to 1. Each value was then vectorized by sampling randomly around the chosen number. The second label is created by randomly selecting numbers that are each less than 0.15, which are also vectorized. The input per sample is 20 vectors with corresponding labels of 0 or 1.

requires a diverse range of information to reach a clinical decision. We consider two classes: (1) 20 random feature values that sum up to 1.0, and, (2) 20 random feature values that are each less than 0.15. These classes were created such that the correct label can only be inferred after inspecting all features. For example, 0.14 is less than 0.15, but it could also be a value that adds to 1 (seen in the last column of Figure 2). If 0.10 was the threshold, the mean of the 20 values would be 0.05, and thus, the sum would also be very close to 1, on average. This would render the task too difficult, and there would not be a significant difference between the samples across the two labels. Setting the threshold to 0.2 would render the task too easy, as on average, the numbers are consistently greater in the second class and the classes could be differentiated using only one modality. Thus, we chose 0.15 as the threshold. Each value was then vectorized by sampling randomly around the selected number, such that each modality is a vector of size 20 rather than a single number, leading to a combined total of 400 features. Overall, the dataset contains 2,000 samples (1,000 for each class). Our constructed simulation dataset tests the scaling capabilities of our method to the extent that real-world datasets could not reach. Figure 2 illustrates the simulation data setting.

## 4.2 REAL-WORLD DATASETS

### 4.2.1 HATEFUL MEMES CHALLENGE DATASET

The Hateful Memes Challenge (14) is an important multimodal dataset to identify hate speech. Originally, the dataset consisted of 10,000 images with associated text annotated with various types of hate speech, but since the original test labels are kept proprietary, we can only use 9,000 samples. The memes were selected in such a way that strictly unimodal classifiers would struggle to classify them correctly. The two modalities are images and text, and the task is to classify a meme as hateful or not. The purpose of using a two-modality dataset is to ensure there is no drop in performance in *all* multimodal scenarios, not just ones where the scalability is significant. The field of multimodal learning has primarily used the attention formula that is consistent with Transformers (43), but we want to highlight that there are other attention formulas (such as (20)) that are effective for multimodal integration.

### 4.2.2 AMAZON REVIEWS DATASET

The Amazon customer review dataset (27) aims to understand consumer opinions and experiences with products sold on Amazon. We sampled from the Electronics category, as it was one of the largest and most commonly used (17; 12; 28). We used three modalities from the dataset – review images, text, and the metadata (price, brand name, product category, etc.) associated with each review. The

task is to classify the binary sentiment of each review - positive (4-5 stars) or negative (1-2 stars), with a total of around 20,000 samples. While sentiment classification with the Amazon reviews data is well-studied unimodally through text (12; 17; 15; 28), we offer a unique multimodal approach to the task. Existing multimodal approaches on Amazon reviews focus on review helpfulness (11; 18), or product similarity (5) rather than sentiment classification. Furthermore, other papers use only text and images or only images and metadata (as text); an image-text-tabular multimodal task has yet to be explored. Intuitively, images and metadata can both be useful for review sentiment prediction - if the image highlights a broken item with the metadata showing a high price or trusted brand, consumer dissatisfaction may be implied.

### 4.2.3   THE CANCER GENOME ATLAS (TCGA) DATASET

TCGA is a cancer genomics program that molecularly characterized over 20,000 primary cancers and matched normal samples spanning 33 cancer types. Accurate prediction of cancer type is an important task but has primarily focused on single-modality approaches (9; 34; 7; 22). Sun et el. (39) used multiple modalities from TCGA but did so for survival prediction rather than cancer classification. We used patients diagnosed with lung (26), stomach (25), colon (24), liver (2), or kidney (44) cancer to create our dataset consisting of 5 modalities – whole-slide images, clinical data (tabular), copy number variation or CNV (tabular), gene expression (tabular), and DNA Methylation (tabular). CNV, DNA Methylation, and gene expression all describe different genomic information; full descriptions can be found in Appendix B. Clinical data includes demographics, laboratory tests, and family relationships. Imaging includes pathology slide images of tissue sampled from the tumor. In total, we had 338 colon cancer patients, 329 kidney, 301 lung, 228 liver, and 226 stomach patients after filtering described in Appendix B. This is a five-class classification task with five modalities.

### 4.3   BASELINES

Our multimodal baselines include a conventional concatenation fusion with no attention, early fusion followed by self-attention, and pairwise cross-attention fusion. The architectures of all models are identical except for the integration stage. For example, since modality-specific encoders can produce different dimension sizes, we add a linear layer before integration to create the same input dimensions. While this step is not strictly necessary for concatenation, we still add the layer there so that no additional factors influence computation costs and performance. While there are many multimodal Transformers available for the vision-language domain, our focus is on examining the underlying fusion mechanism and creating a general integration paradigm for any application, including ones outside of vision-language.

### 4.4   IMPLEMENTATION DETAILS

In the Hateful Memes Dataset, the competition creators pre-determined the validation set and the train set, but the test set labels were not made publicly available. Thus, we created our own test set by randomly sampling 1000 memes from the training set, matching the size of the original competition test set. In all other datasets, for consistency, we randomly sampled 80% of the data for the training set and 10% each for test and validation sets, as there was not an established public split. Our hyperparameter tuning scheme was consistent for each dataset and each model. For each experiment, we used validation accuracy to determine the best parameters. Please see details in Appendix C on exact numbers used to tune for the number of attention heads, learning rate, and number of training steps and Appendix D for compute times and GPU details. We randomly picked 10 random seeds for every experiment - once the best hyperparameters were picked, ten models initialized with those seeds and parameters were run. Then, using the trained 10 models, we evaluated on the test set and took the average of the 10 along with the standard deviation, which is reported in Section 5. To evaluate our model against other integration techniques, we use accuracy and F1-score as measures of performance and the number of floating-point operations (FLOPs) as the measure of runtime complexity. In real-world datasets, FLOPs were measured per sample and reported as the difference between concatenation and multimodal attention ($\Delta$FLOPs).

## 5 RESULTS

### 5.1 SIMULATION RESULTS

Using 2, 5, 10, 15, and 20 simulated modalities, we examine the performance and computation cost across the three integration methods. Most notably, while self-attention and cross-attention grow quadratically with respect to the number of modalities, $k$, ($\mathcal{O}(k^2 \cdot n^2 \cdot d)$), our method scales linearly ($\mathcal{O}(k \cdot n^2 \cdot d)$), as shown in Figure 3 (a). Furthermore, incorporating all 20 modalities makes the task trivial for our method and concatenation. However, cross-attention requires 20 pairwise calculations, which do not help with the classification task (e.g., the local interaction between modality 1 and modality 2 has no impact on the label), thus showing a slight performance drop (Figure 3 (b)). The reported results are averaged across 10 random seeds, demonstrating that OvO attention has less performance variability than cross-attention.

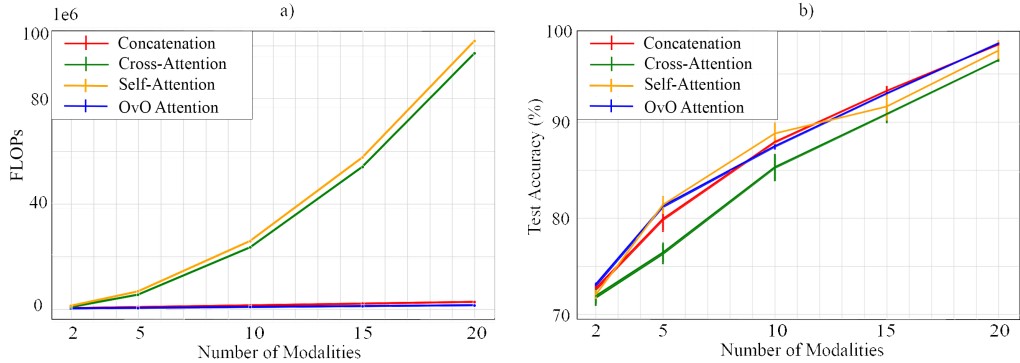

Figure 3: **The impact of using OvO attention to fuse simulated data**. (a) Compute measured in FLOPs and (b) performance measured in accuracy as a function of increasing the number of simulated modalities. OvO attention improves performance at a lower computational cost.

### 5.2 RESULTS ON REAL-WORLD DATASETS

Using three diverse real-world datasets, both in terms of the number of modalities, application domains, and classification tasks, we show that our method consistently improves performance compared to early fusion and pairwise fusion while decreasing computational cost. For the Hateful Memes and Amazon reviews tasks, we used pre-trained models for unimodal text and image classification (BERT and ResNet). In the Amazon reviews case, we used a multi-layer perceptron (MLP) as our unimodal model for the tabular metadata. The same unimodal model architecture was used to encode each modality in the multimodal models. Thus demonstrating how our method provides a more efficient way to integrate modalities than using existing state-of-the-art methods. We perform significance testing between OvO attention's and the next best performing model's accuracy and F1-score means, detailed in the Appendix E. The Hateful Memes results are shown in Table 2 and demonstrate that our model performed better than the baselines, offering a statistically significant improvement in performance to cross-attention (p-value < 0.01) and a slight improvement in FLOPs. This is reasonable as we do not offer substantial scalability gains in the two-modality setting (see Equation 5).

The Amazon Reviews results are shown in Table 3 and demonstrate both performance and scalability advantages of OvO attention. Since the textual modality is most valuable in sentiment prediction, the performance of BERT alone is higher than concatenation, self-attention, and cross-attention. This indicates that the noise from metadata and images interferes with model performance. However, OvO attention can extract information from the other two modalities for a significant performance increase rather than a decrease (p-value < 0.01). Lastly, TCGA results are shown in Table 4. While here, cross-attention had a slightly better performance than OvO, our model offers substantial scalability benefits and, across 10 random seeds, performed consistently, showing robustness and stability.

Table 2: **Hateful Memes results**. Modalities are Image (I) and Text (T). We report the average of 10 random seeds for accuracy, F1-scores, and standard deviations. (*) FLOPs were measured per sample and reported as the difference between concatenation and multimodal attention. We offer improved performance compared to cross-attention across all metrics.

| Model | Modalities | ↑ Accuracy | ↑ F1-Score | ↓ Δ FLOPs |
|---|---|---|---|---|
| ResNet | I | 60.3 ±1.77 | 56.3 ±1.67 | - |
| BERT | T | 66.0 ±1.15 | 61.8 ±1.75 | - |
| Concatenation | I, T | 68.8 ±1.03 | 64.8 ±0.98 | * |
| Self-Attention | I, T | 68.2 ±0.79 | 64.9 ±0.92 | $1.37 \times 10^6$ |
| Cross-Attention | I, T | 69.1 ±0.57 | 65.4 ±0.72 | $1.44 \times 10^6$ |
| **OvO Attention** | **I, T** | **70.7 ±0.87** | **67.7 ±0.97** | **$1.05 \times 10^6$** |

Table 3: **Amazon Reviews results**. Modalities are Image (I), Text (T), and Tabular (Tb). We report the average of 10 random seeds for accuracy, F1-scores, and standard deviations. (*) FLOPs were measured per sample and reported as the difference between concatenation and multimodal attention. We outperform cross-attention across all metrics and offer significant scalability gains.

| Model | Modalities | ↑ Accuracy | ↑ F1-Score | ↓ Δ FLOPs |
|---|---|---|---|---|
| Neural Network | Tb | 57.6 ±0.70 | 57.5 ±0.75 | - |
| ResNet | I | 66.3 ±0.67 | 66.6 ±0.64 | - |
| BERT | T | 92.6 ±0.52 | 92.9 ±0.39 | - |
| Concatenation | I, T, Tb | 92.2 ±0.42 | 92.8 ±0.25 | * |
| Self-Attention | I, T, Tb | 92.4 ±0.39 | 92.4 ±0.39 | $1.90 \times 10^6$ |
| Cross-Attention | I, T, Tb | 91.6 ±0.70 | 92.2 ±0.63 | $1.90 \times 10^6$ |
| **OvO Attention** | **I, T, Tb** | **93.1 ±0.30** | **93.0 ±0.31** | **$0.52 \times 10^6$** |

Furthermore, the difference between cross-attention and OvO attention accuracy and F1-scores was not statistically significant (p-value > 0.01).

Table 4: **TCGA results**. Modalities are Gene Expression (GE), Image (I), Clinical (C), Copy Number Variation (CNV), and DNA Methylation (M). We report the average of 10 random seeds for accuracy and F1-scores along with standard deviations. (*) FLOPs were measured per sample and reported as the difference between concatenation and multimodal attention. We perform competitively compared to cross-attention across all metrics and notably offer more stability as OvO had much lower variation across the seeds (at the two decimals level).

| Model | Modalities | ↑ Accuracy | ↑ F1-Score | ↓ Δ FLOPs |
|---|---|---|---|---|
| Convolutional Neural Network | I | 56.7 ±2.99 | 55.6 ±4.03 | - |
| Neural Network | C | 60.8 ±0.47 | 57.2 ±0.26 | - |
| Neural Network | CNV | 93.4 ±0.80 | 94.1 ±0.74 | - |
| Neural Network | M | 97.0 ±0.43 | 97.4 ±0.36 | - |
| Neural Network | GE | 97.6 ±1.19 | 97.7 ±1.00 | - |
| Concatenation | GE, I, C, CNV, M | 97.8 ±1.50 | 97.9 ±1.26 | * |
| Self-Attention | GE, I, C, CNV, M | 96.8 ±0.96 | 97.0 ±0.89 | $6.56 \times 10^6$ |
| **Cross-Attention** | **GE, I, C, CNV, M** | **99.2 ±1.04** | **99.3 ±0.92** | $5.26 \times 10^6$ |
| **OvO Attention** | GE, I, C, CNV, M | 98.3 ±0.07 | 98.4 ±0.06 | **$0.33 \times 10^6$** |

## 5.3 CASE STUDY: DETERMINING MODALITY IMPORTANCE USING ATTENTION VECTORS

Our model, OvO, generates a single attention vector per modality, allowing us to discern which modality receives the highest attention score and provide insight into the model's decision-making process. Using the TCGA dataset, where OvO was highly successfull, we demonstrated the transparency of OvO attention by investigating the attention vectors on this task (see Appendix F for the heatmap of the Amazon reviews dataset). As shown in Figure 4, each attention context vector is computed by averaging across the embedding dimension and the 10 random seeds used for our best model. The horizontal axis enumerates the test set patients, while the vertical axis denotes

the "main" modality from the attention score (the modality not inside the average).In Figure 4, we observed the gene expression attention vector as the highest scored, corroborated by Table 4 where gene expression emerged as the highest performing single modality. Although DNA Methylation performs well unimodally, it isn't emphasized in the heatmap due to its established strong biological correlation with gene expression (45; 3; 35), avoiding redundancy. Similarly, the clinical modality's low performance is reflected in the attention heatmap, while imaging, the only non-tabular modality, provides new information to the model and thus highlighted. Overall, the construction of OvO offers a more interpretable framework than self-attention and cross-attention.

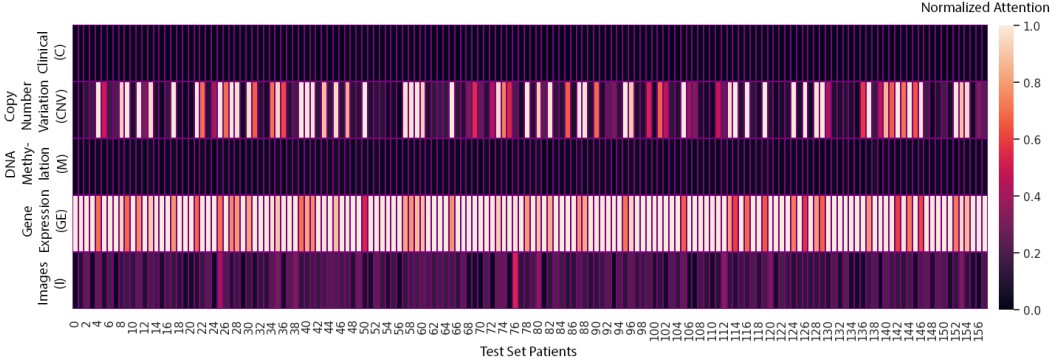

Figure 4: **Attention heatmap for the TCGA dataset**. Each attention vector is computed by averaging across the embedding dimension and the 10 random seeds used to report our best model. The x-axis includes the patients from the test set, and the y-axis includes the "main" modality from the attention context vector. The attention weights are Gene Expression (GE) vs. others, Image (I) vs. others, Clinical (C) vs. others, Copy Number Variation (CNV) vs. others, and DNA Methylation (M) vs. others. This figure is consistent with our understanding of the biological relationship between gene expression and DNA Methylation as well as our single-modality results from Table 4.

## 6 DISCUSSION

Our proposed method, OvO, provides a way to overcome one of the major challenges associated with multimodal datasets - computational resource demand and cost. OvO is also more interpretable, enabling adoption in high-stakes settings, such as clinical decision support.

While many deep learning studies utilize datasets with millions of samples, we were limited by our computational resources. We argue that the primary difference between our approach from early fusion and pairwise fusion is not about the number of samples but the number of modalities. To support this claim, we demonstrated the efficiency of OvO on 20 modalities in our simulation and conducted a complexity analysis. We believe that our results, based on the resources available, give a robust showcase of the main concept of our work.

The literature on Transformers and, specifically, Multimodal Transformers provides a long list of architectures, some of which make efforts towards scalability. However, such models focus on the vision-language domain, and are not applicable in all multimodal scenarios (e.g., clinical decision support). Thus, we chose three common fusion paradigms (similar to (23)) that encapsulate most of the underlying attention mechanisms in the multimodal literature. We aim to modify the basic attention mechanism in such models to make multimodal fusion more scalable with an increasing number of modalities.

## 7 CONCLUSION

We present One-Versus-Others (OvO), a new scalable multimodal attention mechanism. The proposed formulation involves averaging the weights from each modality during training, significantly reducing the computational complexity compared to early fusion through self-attention and cross-attention methods. OvO outperformed self-attention, cross-attention, and concatenation on three diverse real-world datasets and on a simulation dataset that shows the scalability gains in an extremely multimodal setting. The results demonstrate that the proposed approach improves performance compared to state-of-the-art fusion techniques while decreasing computation costs.

## 8 REPRODUCIBILITY STATEMENT

Alongside this paper, we provide the source code for our work[1]. The repository contains the implementation details of all of our experiments as well as a requirements file to install the necessary libraries and a ReadMe.md file containing running instructions to reproduce our results. In the text, we describe implementation details in Section 4.4 and provide details in Appendix C on exact numbers used to tune for the number of attention heads, learning rate, and number of training steps. In Appendix D, we describe the compute times and GPU requirements used in our work. Data preprocessing steps are also included in the source code, as well as in Appendix B. Lastly, Appendix Section A provides step-by-step details for the complexity calculations shown in Section 3.3.

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

# A    COMPUTATIONAL COMPLEXITY ANALYSIS FOR MULTIMODAL INTEGRATION SCHEMES

In this section, we present the step-by-step details of the computational complexity analysis presented in Section 3.3. The analysis is done with respect to the size of the input modalities associated with the three paradigms used in our experimental setting: early fusion followed by self-attention, cross-modal attention, and One-Versus-Others (OvO) Attention.

## A.1    EARLY FUSION

The early fusion approach involves first combining the modalities and then processing the concatenated sequence with the self-attention mechanism.

**Step 1: Concatenation of Modalities.**

Let $k$ be the number of modalities and $n$ be the feature-length of each modality.

$$\text{Total length after concatenation} = k \times n$$

The complexity for this operation is linear:

$$\mathcal{O}(k \cdot n)$$

**Step 2: Compute Queries, Keys, and Values.**

The self-attention mechanism derives queries (Q), keys (K), and values (V) for the concatenated sequence (length $k \cdot n$) using linear transformations with representation dimension, $d$. The complexity of each transformation operation is:

$$\mathcal{O}(k \cdot n \cdot d)$$

**Step 3: Compute Attention Scores.**

Attention scores are computed by taking the dot product of queries and keys. The self-attention mechanism has quadratic complexity with respect to the sequence length and linear complexity with respect to the representation dimension $d$ (43). Thus, given the concatenated sequence's length of $k \cdot n$ and the dimension of the keys and queries $d$, the complexity of this step is:

$$\mathcal{O}((k \cdot n)^2 \cdot d) = \mathcal{O}(k^2 \cdot n^2 \cdot d)$$

**Step 4: Calculate the Weighted Sum for Outputs.**

For each of the $k \cdot n$ positions in the concatenated sequence, we compute the softmax of the attention scores to produce the attention weights. These weights are then multiplied with their corresponding $d$-dimensional values to compute the weighted sum, which becomes the output. The computational complexity of these operations is:

$$\mathcal{O}(k^2 \cdot n^2 \cdot d)$$

When combining all steps, the dominating terms in the computational complexity stem from the attention scores' computation and the weighted sum, culminating in an overall complexity of:

$$\mathcal{O}(k^2 \cdot n^2 \cdot d)$$

## A.2    CROSS-MODAL ATTENTION

For cross-modal attention, each modality attends to every other modality.

**Step 1: Compute Queries, Keys, and Values for Inter-Modal Attention.**

From a given modality, compute a query (Q), and from the remaining $k - 1$ modalities, compute keys (K) and values (V). Keys, queries, and values are obtained using linear transformations with representation dimension $d$. The complexity of each transformation operation is:

$$\mathcal{O}(n \cdot d) \text{ for each query, key, value set}$$

Considering all modalities:

$$\mathcal{O}(k \cdot (k-1) \cdot n \cdot d)$$

The term $k \cdot (k-1)$ comes from the number of pairwise permutations of $k$, given by $_kP_2 = \frac{k!}{(k-2)!} = k(k-1)$.

**Step 2: Calculate Attention Scores for Inter-Modal Attention.**
The queries and keys from different modalities are used to compute attention scores, which represent how much one modality should attend to another.

$$\mathcal{O}(n^2 \cdot d) \text{ for each pair of modalities} \quad (43)$$

Considering all modalities:

$$\mathcal{O}(k \cdot (k-1) \cdot n^2 \cdot d)$$

**Step 3: Calculate the Weighted Sum for Outputs.**
For every modality interaction, calculate the softmax of the attention scores to obtain the attention weights. These weights are then used in conjunction with the values vector to derive the weighted sum for the output:

$$\mathcal{O}(n^2 \cdot d) \text{ for each pair of modalities}$$

Considering all modalities:

$$\mathcal{O}(k \cdot (k-1) \cdot n^2 \cdot d)$$

When evaluating all steps together, the dominating factors in computational complexity arise from the computation of attention scores and the weighted sum. Thus, the collective complexity for cross-modal attention, where each modality attends to every other, equates to:

$$\mathcal{O}(k \cdot (k-1) \cdot n^2 \cdot d) = \mathcal{O}((k^2 - k) \cdot n^2 \cdot d)$$

For the complexity of cross-modal attention, the dominant term is $k^2$. The $k-1$ term effectively becomes a constant factor in relation to $k^2$. As $k$ tends toward larger values, the difference between $k^2$ and $k^2 - k$ diminishes. This is a consequence of the principles of big $O$ notation, which focuses on the fastest-growing term in the equation while dismissing constant factors and lower-order terms. As a result, for asymptotic analysis, the complexity

$$\mathcal{O}(k^2 - k) \cdot n^2 \cdot d$$

can be simplified to:

$$\mathcal{O}(k^2 \cdot n^2 \cdot d)$$

.

### A.3 One-Versus-Others (OvO) Attention Complexity

**Step 1: Averaging of "Other" Modalities.**
Let $k$ be the number of modalities and $n$ be the feature-length of each modality. For each modality $m_i$, averaging over the other $k-1$ modalities results in a complexity of:

$$\mathcal{O}(n)$$

Given that this needs to be computed for all $k$ modalities:

$$\mathcal{O}(k \cdot n)$$

**Step 2: Calculate Attention Scores with Shared Weight Matrix W.**
The modality vector $m_i$ and the average of "other" modalities, $\frac{\sum_{j \neq i}^n m_j}{n-1}$, are used to compute attention scores, which represent how much one modality should attend to the others. Multiplication with the weight matrix $W$ (with representation dimension $d$) and the dot product with the summed modalities lead to:

$$\mathcal{O}(n^2 \cdot d)$$

Considering this operation for all $k$ modalities:

$$\mathcal{O}(k \cdot n^2 \cdot d)$$

**Step 3: Calculate the Weighted Sum for Outputs.**
For every modality interaction, calculate the softmax of the attention scores to obtain the attention weights. These weights are then used in conjunction with the $m_i$ vector (analogous the values (V) vector) to derive the weighted sum for the output:

$$\mathcal{O}(n^2 \cdot d) \text{ for each pair of modalities}$$

Considering all modalities:

$$\mathcal{O}(k \cdot n^2 \cdot d)$$

When evaluating all steps together, the dominating factors in computational complexity arise from the computation of attention scores. Thus, the collective complexity for cross-modal attention, where each modality attends to every other, equates to:

$$\mathcal{O}(k \cdot n^2 \cdot d)$$

In summary, One-Versus-Others (OvO) Attention exhibits a computational complexity that grows linearly with respect to the number of modalities ($\mathcal{O}(k \cdot n^2 \cdot d)$). In contrast, both early fusion through self-attention and cross-attention approaches demonstrate quadratic growth with respect to the number of modalities ($\mathcal{O}(k^2 \cdot n^2 \cdot d)$). This makes OvO a more scalable option for multimodal integration.

## B  TCGA MODALITY DESCRIPTIONS AND DETAILED PRE-PROCESSING

CNV defines the varying number of repeats of genetic fragments found in a human genome. The number of repeats of specific genetic fragments influences gene expression levels and has been associated with the progression of different cancers (32). Any genomic regions missing CNV values or only having one unique value across all cases were removed. DNA methylation represents the amount of condensation of genetic regions due to the chemical alteration imposed by methyl groups. This condensation generally represses gene activity near the genetic region. Any genomic regions with missing values were removed. Clinical data includes information such as the patient's diagnosis, demographics, laboratory tests, and family relationships. Categorical features were isolated and a coefficient of variation test was run to determine highly variable features. Features with a coefficient of variation higher than 70 were kept for analysis, along with the target variable. These features were converted into numerical format using one-hot-encoding. Gene expression data is collected through RNA-sequencing. Levels of gene expression are recorded by detecting the amounts of transcripts found for each gene. These levels can be used to determine the molecular mechanisms underlying cancer. Transcriptomic data was filtered to only include protein-coding genes and measured in fragments per kilobase of exon per million mapped fragments (FPKM). Imaging - TCGA collects pathology slide images of tissues sampled from the tumor. This modality provides visual information about the malignant region and can help with diagnosis and treatment planning. The image data was filtered only to include DX images, which result from a single X-Ray exposure, rotated to landscape view, then cropped to the median aspect ratio of 1.3565. We filtered for patients that had all five modalities, and we also only chose the patients that were still alive, to create a more balanced number of patients between cancer types (338 colon cancer patients, 329 kidney, 301 lung, 228 liver, and 226 stomach patients, after the filtering). The task we created is to classify each patient's cancer type. For all modalities, features with missing values were dropped. For CNV, DNA Methylation, and gene expression data, feature reduction was performed using a random forest classifier, only on training data, ensuring the test was not seen by the random forest. Using the validation set, we determined the best number of estimators (out of 50, 100, 150).

## C  HYPERPARAMETER TUNING

For each experiment, we used the validation accuracy to determine the best hyperparameters. We tuned the learning rate (0.01 - $1 * 10^{-8}$), batch size (16, 32, 64, 128), epochs (200 epochs with early

stopping if validation accuracy did not increase for 5 epochs), and number of attention heads for the OvO and pairwise cross-modal attention models (1, 2, 4, 8, 16). For the neural network encoders, we tuned the number of linear layers ranging from 1 to 4. Similarly, for the convolutional neural network, we tuned the number of convolution layers ranging from 1 to 4.

## D  COMPUTE RESOURCES

For each experiment, we use one NVIDIA GeForce RTX 3090 GPU. For the Hateful Memes task, single-modality models ran for roughly 40 minutes, and multi-modal models ran for roughly 55 minutes on average. For the Amazon reviews task, the single modality pre-trained models ran for roughly 50 minutes, the single modality neural network ran for a minute, and the multi-modal models ran for approximately an hour on average. For the TCGA task, single-modality models ran for 5 minutes, while multi-modal models ran for roughly 15 minutes on average. In the simulation dataset, the maximum modalities was 20 which took our model, OvO, roughly 2 minutes to run, while the cross-modal attention baseline took about 20 minutes to run on average.

## E  SIGNIFICANCE TESTING

We use a t-test to determine if there is a significant difference in accuracy and F1-score means between OvO attention and the next best-performing multimodal model. Our sample size is 10 from each group, as we initialized the models with 10 random seeds. For the Hateful Memes dataset, we compare against cross-attention as it performed the second best after OvO. Using an $\alpha = 0.01$, we have evidence to reject the null hypothesis and conclude that there is a statistically significant difference in means between cross-attention and OvO attention. The p-value for the accuracy scores is $1.22e^{-8}$ and the p-value for F1-scores is $1.89e^{-5}$. For the Amazon reviews dataset, we compare against self-attention as it performed the second best after OvO. We get a p-value for accuracy scores of $4.77e^{-4}$ and a p-value of $4.87e^{-4}$ for F1-scores. Thus, we demonstrate a statistically significant difference in accuracy and F1-score means between self-attention and OvO attention. Lastly, for the TCGA dataset, we do not have evidence to reject the null hypothesis and cannot say that the accuracy and F1-score means were different between OvO and cross-attention since the p-values were greater than $\alpha = 0.01$ (p-value of 0.04 for accuracy means, and p-value of 0.02 for F1-score means). This demonstrates that although cross-attention performed slightly better than OvO, it was not statistically significant.

## F  CASE STUDY FOR ATTENTION VECTORS ON AMAZON REVIEWS DATASET

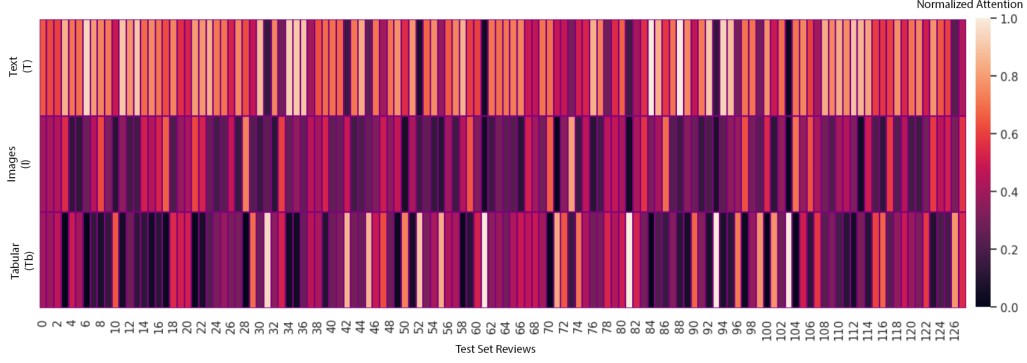

Figure 5: **Attention heatmap for Amazon reviews dataset**. Each attention vector is computed by averaging across the embedding dimension and across the 10 random seeds used to report our best model. The horizontal axis includes a sample of size 128 (batch size of the model) reviews from the test set, and the vertical axis includes the "main" modality from the attention score. The attention scores Text (T) vs. others, Images (I) vs. others, and Tabular (Tb) vs. others. This figure is consistent with the single-modality results from table 3.

Since our model, OvO, performed well on the Amazon reviews task and could be used for future sentiment analysis tasks reliably, we wanted to explore the attention scores on this task. Each attention context vector shown in Figure 5 is computed by averaging across the embedding dimension and across the 10 random seeds used to report our best model. The X-axis includes a sample of size 128 (batch size of the model) reviews from the test set and the y-axis includes the "main" modality from the attention score. The attention weights are Text (T) vs. others, Images (I) vs. others, and Tabular (Tb) vs. others. We observe that the text attention vector is the most highly scored, which is supported by the single-modality results from Table 3, where text was the highest performing single modality. Thus, we further demonstrate that OvO can be used to better understand modality importance, without the need for ablation studies. This is significant because knowing which modality is most important to decision-making can motivate future data collection efforts in diverse research environments and help make deep learning models more transparent.

