# OpenReview forum: "One-Versus-Others Attention: Scalable Multimodal Integration"
_ICLR.cc/2024/Conference — ICLR 2024 Conference Withdrawn Submission_

### Official Review · Reviewer_YfMJ · 2023-10-20

**Soundness:** 3 good
**Presentation:** 3 good
**Contribution:** 3 good
**Rating:** 5
**Confidence:** 3

**Summary:**

The paper presents One-Versus-Others (OvO), a new scalable multimodal attention mechanism. The proposed formulation involves averaging the weights from each modality during training, significantly reducing the computational complexity compared to early fusion through self-attention and cross-attention methods. OvO outperformed self-attention, cross-attention, and concatenation on three diverse real-world datasets and on a simulation dataset that shows the scalability gains in an extremely multimodal setting. The results demonstrate that the proposed approach improves performance compared to state-of-the-art fusion techniques while decreasing computation costs.

**Strengths:**

+ The paper presents, OvO, a generalizable multimodal integration scheme that is domain-neutral and does not require modality alignment;
+ OvO scales linearly with the number of modalities, while also performing competitively to self-attention and cross-attention;
+ The paper performs robust benchmarking on new simulated and real-world multimodal integration tasks.

**Weaknesses:**

- One of the major weaknesses of the paper is that the experiment section is not convincing. The datasets used are simulation dataset and small scale datasets or datasets which are not reported by other compared methods such as VisualBERT, VL-BERT, etc. The main argument of the paper is that, the paper proposes a scalable one-versus-others attention, which is better than cross-attention used in LXMERT and ViLBERT and self-attention used in VisualBERT and VL-BERT. Thus a fair comparison would be conducting experiments on the same datasets reported by these methods.

- The multimodal fusion strategy is also explored in unified content code extraction of multimodal generation [1]. Adding related work in the reference could make the paper more smooth and have bigger impact.
[1] Multi-Domain Image Completion for Random Missing Input Data, IEEE Transactions on Medical Imaging, 2021.

**Questions:**

Please check Weaknesses

---

> ### Author Response · Authors · 2023-11-18
>
> We thank the reviewer for the thoughtful feedback. In what follows, we provide responses to the main weaknesses.
>
> **Weaknesses:**
> 1. “One of the major weaknesses of the paper is that the experiment section is not convincing. The datasets used are simulation dataset and small scale datasets or datasets which are not reported by other compared methods such as VisualBERT, VL-BERT, etc. The main argument of the paper is that, the paper proposes a scalable one-versus-others attention, which is better than cross-attention used in LXMERT and ViLBERT and self-attention used in VisualBERT and VL-BERT. Thus a fair comparison would be conducting experiments on the same datasets reported by these methods.”
> We thank the reviewer for their comment and understand their concerns. While models such as LXMERT, ViLBERT, VisualBERT, and VL-BERT, are popular and do train on large-scale datasets, they are limited to two modalities (images and text). Our work aims to find a general method that can be applied in multimodal settings that expand beyond two modalities and can be used in any research environment (with less computational costs). We, too, were restricted by our computational resources and thus could not use the large-scale datasets that the aforementioned models were trained on. Our method demonstrates efficiency with a growing number of modalities, which is why we focused on a wider range of modalities: 2-modality data (Hateful Memes), 3-modality data (Amazon Reviews), 5-modality data (TCGA), and 20-modality data (simulation).
>
> 2. “The multimodal fusion strategy is also explored in unified content code extraction of multimodal generation [1]. Adding related work in the reference could make the paper more smooth and have bigger impact. [1] Multi-Domain Image Completion for Random Missing Input Data, IEEE Transactions on Medical Imaging, 2021.”
> We thank the reviewer for this suggestion and add this work and reference to the Related Works.

---

> > ### Comment · Reviewer_YfMJ · 2023-12-03
> > **Thank you so much for the response!**
> >
> > Thank you so much for taking time writing the response! The response has been carefully reviewed. I personally totally understand the computation resources cost concern especially for authors in the academies. However, after careful consideration about other reviewers' comments and competitive results in other submissions, the rating is tuned a little bit. The paper is pretty novel and high impact. Hope the authors could continue working on it based on the reviews. Thank you!

---

### Official Review · Reviewer_UUiq · 2023-10-31

**Soundness:** 2 fair
**Presentation:** 2 fair
**Contribution:** 2 fair
**Rating:** 5
**Confidence:** 5

**Summary:**

The current state of multimodal learning, particularly in the medical field, is confronted with challenges stemming from the diverse nature of data inputs, such as X-rays and PET scans. These varied data types necessitate a method for efficient and precise information integration. In response to this, the authors have introduced an innovative attention mechanism, termed "one-versus-others." This mechanism stands out for its ability to scale linearly with the number of input modalities, presenting a significant advantage in handling multimodal data. The effectiveness of this approach has been validated through rigorous testing on three real-world datasets, where it consistently outperformed other existing fusion techniques, showcasing its potential to enhance performance in multimodal learning applications.

**Strengths:**

+ The authors have introduced "one-versus-others," a versatile and scalable approach for integrating multimodal data without the need for modality alignment.
+ Despite its linear scalability with the increasing number of modalities, this method competes effectively with other attention mechanisms in terms of performance.
+ Demonstrating robust applicability, "one-versus-others" has shown promising results on both simulated and real-world multimodal integration tasks, indicating its potential as a reliable tool for handling diverse data inputs.

**Weaknesses:**

- The linear time complexity of "one-versus-others" is commendable; however, its storage requirements are substantial. This is evident from Equation 6, where outputs from all attention heads are concatenated and subjected to another multihead attention operation, essentially amounting to a sum of all previous module results.
- The authors highlight the prevalent focus of existing methods on NLP applications, yet their experiments predominantly utilize NLP datasets. To bolster their argument and the generalizability of their method, the inclusion of diverse data inputs from varied domains, such as x-ray or PET scans, would be beneficial.
- The datasets used, Amazon and hateful memes, feature a limited number of modalities. This scenario does not truly challenge or demonstrate the linearity of the proposed method.
- The manuscript appears imbalanced, with the methodology section spanning just one page, and a disproportionate amount of content dedicated to dataset settings and experimental configurations. A recalibration of focus towards the methodological aspects of the paper is suggested.
- The reported improvements in results are modest, with most datasets showing an enhancement of merely 1%. Such marginal gains may not sufficiently underscore the significance of the proposed method.

**Questions:**

- The utilization of a simulated dataset is perplexing. The multimodal setting is not clearly articulated, and the necessity of simulations is questionable, especially if the proposed method is as universally applicable as suggested. The availability of multiple real-world datasets should negate the need for simulated scenarios.
- Equation 3 outlines a weighted averaging approach to calculate attention weights, seemingly derived from prior works. This approach could potentially dilute the informative value of the inputs. It would be beneficial for the authors to delve deeper into this aspect and provide empirical or theoretical insights to address these concerns.

---

> ### Author Response · Authors · 2023-11-18
>
> We thank the reviewer for the thoughtful feedback. In what follows, we provide responses to the main weaknesses and questions.
>
> **Weaknesses:**
> 1. “OvO’s storage requirements are substantial (Eq 6).”
> We would like to clarify that Eq 6 is derived from the "Attention is All You Need paper" and does not add any computational burden to multiheaded attention. The equation from the original multiheaded attention is:
> $$
> \text{MultiHead}(Q, K, V) = \text{concat}(\text{head}_1, \ldots, \text{head}_h)W^O
> $$
> where
> $$
> \text{head}_i = \text{Attention}(QW^Q_i, KW^K_i, VW^V_i)
> $$
> OvO multiheaded attention:
> $$
> \text{MultiheadedOvOAttention}(m_i, \{m_j: j \neq i\}) = \text{concat}(\text{head}_1, \ldots, \text{head}_h)W^O
> $$
> where
> $$
> \text{head}_k = \text{OvO Attention}(m_iW_k^{m_i}, \{m_jW_k^{m_j}: j \neq i\})
> $$
> Storage-wise, OvO and the baselines are similar, as all models share identical parameters except in the fusion stage. For instance, each model needs ~487 MB of storage for the Amazon Reviews data, regardless of the fusion approach.
> 2. “The experiments predominantly utilize NLP datasets.”
> We direct the reviewer’s attention to Section 4.2.3 and Table 4, which discuss the five-modality medical datasets used in our experiments. We agree with the reviewer that the generalizability of the method is stronger on diverse data; thus, we have used The Cancer Genome Atlas (TCGA) in our work. TCGA’s modalities are whole-slide images, clinical data, CNV, gene expression, and DNA Methylation (Section 4.2.3 and Appendix B). Although PET scans or X-rays weren't used, whole-slide images serve a similar role in showcasing generalizability. The results on TCGA (Table 4), demonstrate OvO’s success on a dataset outside the NLP domain.
> 3. “The datasets used, Amazon and hateful memes, feature a limited number of modalities.”
> We would like to clarify that the Hateful Memes dataset is designed to demonstrate that the OvO formula does not affect the performance negatively for the simple 2-modality case. To demonstrate the linearity of OvO and the generalizability of the method, we present four multimodal scenarios: 2-modality data (Hateful Memes), 3-modality data (Amazon Reviews), 5-modality data (TCGA), and 20-modality data (simulation). We believe that this range of datasets is sufficient to demonstrate the efficiency of OvO, and we are happy to discuss these results further.
> 4. “The manuscript appears imbalanced.”
> We will keep this suggestion in mind and rearrange sections of the paper appropriately.
> 5. “The reported improvements in results are modest, with most datasets showing an enhancement of merely 1%.”
> We recognize the reviewer's concern and emphasize that our work aims to achieve superior computational efficiency rather than surpassing performance of existing methods. Since we do observed performance gains, we use hypothesis testing to show that OvO's increase in accuracy is statistically significant. In terms of percentages, in the Amazon Reviews data, OvO's 523,520 FLOPs represent a 72.50% reduction compared to the 1,903,616 FLOPs each for cross and self-attention when isolating attention layers. In the five-modality TCGA dataset, OvO's 330,240 FLOPs amount to reductions of 93.73% and 94.96% compared to cross and self-attention's 5,260,800 and 6,556,160 FLOPs, respectively, underscoring OvO's superior efficiency.
>
> **Questions:**
> 1. “The utilization of a simulated dataset is perplexing.”
> We demonstrate the generalizability of the proposed formulation on three real-world datasets ranging from 2 to 5 modalities. However, as the reviewer pointed out, the real-world datasets may not have enough modalities to demonstrate linearity in complexity beyond a certain number. Therefore, to systematically study the complexity of OvO attention and to show that it is indeed linear with respect to the number of modalitiles, we designed the simulation to scale the number of modalities to 20. Our results show that as the number of modalities increases, OvO grows linearly, whereas self and cross-attention grow quadratically.
> 2. “Eq 3 outlines a weighted averaging approach to calculate attention weights, that could potentially dilute the informative value of the inputs.”
> We would like to clarify that Eq 3 is not a weighted average, as $m_i$ is not a scalar coefficient but the “main” modality that gets multiplied by all other modalities and a neural network weight matrix (W). This dot product creates a similarity function that is the essence of every attention calculation (Section 3.1). The suspected dilution of information would be evident if there were significant decreases in performance, but we observe the opposite. Specifically, in Section 5.3 and Appendix F, we show attention heatmaps for the TCGA and Amazon Reviews data, that suggest that the strongest unimodal modalities are also the ones paid most attention to. This means that the attention scheme we created is working appropriately and not losing information.

---

> > ### Comment · Reviewer_UUiq · 2023-12-03
> > **Response to Comments**
> >
> > Thank you for your detailed response. I appreciate the efforts made to address the concerns. However, I would like to summarize some comments:
> >
> > - The response did not adequately address the content imbalance, particularly concerning the method section. The methodology is crucial in any research paper and merits comprehensive discussion. I recommend expanding this section to appropriately reflect its significance.
> > - The choice of a simulated dataset remains perplexing, especially since the authors acknowledge the rarity of datasets with a larger number of modalities in their response. This raises questions about the paper's relevance and applicability. I suggest the authors to clarify the necessity and contribution of this approach.
> > - The study highlight the difference between the application of methods from Natural Language Processing (NLP) but then appears to re-apply these methods to NLP tasks. This seems contradictory, and I suggest the authors address this inconsistency in their approach.
> > - Both reviewers (YfMJ and sMs) have expressed concerns about the use of the simulated dataset, indicating a shared skepticism about its inclusion and effectiveness. This warrants a reevaluation of its role in the study.
> > - Reviewer sMs5 highlighted limitations regarding the novelty of the proposed method, primarily that it seems to be an average over each modality in the representation layer. I concur with this observation and recommend that the authors more clearly delineate the innovative aspects of their method.
> >
> > To sum up, It addressed some of my concerns. Thus, I tuned the rating a little bit.

---

### Official Review · Reviewer_sMs5 · 2023-11-01

**Soundness:** 2 fair
**Presentation:** 3 good
**Contribution:** 2 fair
**Rating:** 5
**Confidence:** 4

**Summary:**

This paper proposes a scalable attention mechanism for multimodal fusion named One-Versus-Others (OvO).  OvO averages the weights from each modality during training, reducing the computational complexity compared to early fusion through self-attention and cross-attention methods. The results demonstrate that the proposed approach outperforms self-attention, cross-attention, and concatenation on three diverse real-world datasets and a simulation dataset.

**Strengths:**

1. This paper proposed a multimodal attention fusion mechanism, scaling linearly with the number of modalities, which is more efficient than self-attention and cross-attention.
2. The proposed OvO has the potential for a large number of modalities due to the small computational costs and compared performance.

**Weaknesses:**

1. The novelty seems limited, as this paper's contribution is only a new design of multimodal attention that averages weights from all other modalities’ encoders.
2. The introduction of the baseline model is insufficient,  including the architecture details and the references.
3. In the experimental results of the simulated dataset in Figure 3, it seems that the performance improvement achieved by OvO is marginal compared to concatenation with similar FLOPs.
4. The compared method is insufficient. There are some different fusion mechanisms in  [1] such as Hierarchical Attention.
    * [1] Xu P, Zhu X, Clifton D A. Multimodal learning with transformers: A survey[J]. IEEE Transactions on Pattern Analysis and Machine Intelligence, 2023.

**Questions:**

1. Please compare parameters to demonstrate the efficiency of OvO further.
2. Is there any reference for the simulation of the 20 simulated modalities? Why is the simulation analogous to a medical setting? Please clarify.
3. Is there any pre-trained model used for the training? The self-attention and cross-attention are both used to pretrain large models (e.g.,  ALBEF,  VisualBERT) with large datasets (e.g., COCO[1], SBU Captions[2]). However, the scale of datasets in the paper is relatively small, which is unfair. Could the OvO achieve compared performance on the big dataset? Please clarify the model details and the task application to demonstrate the advantages of the proposed method.
5. Please check the formula  $k^P_2$ in Section 3.3.

    * [1]T.-Y. Lin et al., “Microsoft COCO: Common objects in context,” in Proc. Eur. Conf. Comput. Vis., 2014, pp. 740–755.
    * [2] V. Ordonez, G. Kulkarni, and T. Berg, “Im2Text: Describing images using 1 million captioned photographs,” in Proc. Int. Conf. Neural Inf. Process. Syst., 2011, pp. 1143–1151.

---

> ### Author Response · Authors · 2023-11-18
>
> We thank the reviewer for the thoughtful feedback. In what follows, we provide responses to the main weaknesses and questions.
>
> **Weaknesses:**
> 1. “The contribution is only a new design of multimodal attention that averages weights from all other modalities’ encoders.”
> We appreciate the reviewer’s feedback and clarify that the OvO formula is a different formulation than the one used in self and cross-attention. It is more involved than just averaging weights of all other modalities’ encoders, as it learns similarity by multiplying one modality by a neural network weight matrix W, and then to the average of all other modalities. This step is crucial as W becomes a learned parameter that helps strengthen inter-modal understanding.
> We believe that the contribution of the work should not be judged based on the simplicity of the solution but on the complexity of the problem it solves. OvO offers a new domain-neutral attention mechanism that substantially lowers computational complexity and maintains competitive performance compared to existing methods. This is increasingly relevant in diverse domains like healthcare, e-commerce, and autonomous vehicles, where efficient data integration is key.
> 2. “Baseline descriptions are insufficient.”
> Baseline descriptions are in Section 4.3, with additional dataset specifics in Section 4.2. The baselines are consistent across tasks, and the only differences lie in the modality encoders, like BERT for text and ResNet for images, or a standard multilayer perceptron for medical modalities where there is no known encoder (see Section 5.2). We are happy to make further clarifications.
> 3. “In simulation, the performance improvement achieved by OvO is marginal compared to concatenation.”
> The simulation experiment aims to demonstrate the linear complexity growth with increasing modalities. Figure 3b, showing accuracy, ensures that reduced FLOPs do not mean a drop in performance. While the simulation task was designed to be straightforward for any method, including concatenation, real-world tasks, being more complex, usually favor attention-based models. Hence, in real-world datasets, our method shows a statistically significant improvement over other attention schemes and concatenation.
> 4. “The compared method is insufficient (e.g., Hierarchical Attention).”
> While cross-attention and self-attention are two different uses of the attention formula, hierarchical attention is not a fundamental variation on attention but rather an early fusion model with self-attention in it, followed by a multi-stream scheme, which does not involve further fusion via attention.
>
> **Questions:**
> 1. “Please compare parameters.”
> OvO's efficiency stems from its linear, rather than quadratic, modality fusion. The number of parameters remains consistent across fusion schemes. For example, concatenation, self-attention, cross-attention, and OvO attention, all maintain 122,027,094 parameters for the Amazon Reviews dataset. The parameter count consistency is due to the constant dimensions of input and output layers across fusion methods. In FLOPs, a key metric of computational efficiency in deep learning models, OvO's 523,520 FLOPs significantly undercut the 1,903,616 FLOPs of both cross and self-attention, marking a 72.50% reduction and highlighting OvO's efficiency.
> 2. “Why is the simulation analogous to a medical setting?”
> In medical diagnostics, providers must consider all available information. Neglecting any modality (e.g., disregarding imaging and focusing solely on genetics) can compromise patient assessment and risk misdiagnoses. Since the medical domain is where more modalities are present - imaging alone can span many types (pathology images, MRIs, PET scans, X-rays, etc.), we wanted to simulate a scenario where every simulated modality was needed to classify correctly. The simulation's main purpose is to demonstrate linear complexity growth with increasing modalities. We'll relocate this section to avoid confusion to align with the complexity analysis, not the dataset results.
> 3. “The datasets in the paper are small, which is unfair as self and cross attention are pretrained on large datasets.”
> While models such as ALBEF and VisualBERT train on large-scale datasets, they are limited to two modalities (images and text). We aim to find a general method that can expand beyond two modalities and can be used in any research environment (with less computational costs). We, too, were restricted by our computational resources and could not use the large-scale datasets that the aforementioned models were trained on. Our method demonstrates efficiency with a growing number of modalities, which is why we focused on a wider range of modalities:  2-modality data (Hateful Memes), 3-modality data (Amazon Reviews), 5-modality data (TCGA), and 20-modality data (simulation).
> 4. “Check the formula in Section 3.3.”
> We have corrected the notation in the permutation formula.

---

### Author Response · Authors · 2023-11-18

We would like to thank the reviewers for their valuable feedback on the paper. We were glad to see that the reviewers appreciated the efficiency and scalability of our proposed OvO (One-versus-Others) attention mechanism, particularly its linear scalability with the number of modalities and competitive performance compared to existing self-attention and cross-attention mechanisms. We were also encouraged by the recognition of our robust benchmarking on new simulated and real-world multimodal integration tasks, which highlights the practical applicability and reliability of OvO in diverse scenarios.

We address the concerns raised by the reviewers in individual responses and look forward to a productive discussion with them.